# Intubation with channeled versus non-channeled video laryngoscopes in simulated difficult airway by junior doctors in an out-of-hospital setting: A crossover manikin study

Shi Hao Chew[1,2☺]*, Jonathan Zhao Min Lim[1,2☺], Benjamin Zhao Bin Chin[1,2], Jia Xin Chan[1], Raymond Chern Hwee Siew[3☺]

1 Department of Anaesthesia, National University Hospital, Singapore, 2 Headquarters Army Medical Services, Singapore Armed Forces, Singapore, 3 RS Anaesthesia & Intensive Care, Singapore

☺ These authors contributed equally to this work.
* shi_hao_chew@nuhs.edu.sg

## Abstract

Failure to secure the airway is an important cause of morbidity and mortality during resuscitations. We compared the rate of successful intubation of the King Vision™ aBlade™ channeled and non-channeled video laryngoscopes, and McGRATH™ MAC video laryngoscope when used by junior doctors to intubate a simulated difficult airway in an out-of-hospital setting. 105 junior doctors were recruited in a crossover study to perform tracheal intubation with the three video laryngoscopes on a simulated difficult airway using the SimMan® 3G manikin. Primary outcome was the rate of successful intubations. Secondary outcomes were time-to-visualization, time-to-intubation and ease of use. Rates of successful intubations were higher for King Vision channeled and McGrath compared to the King Vision non-channeled (85.7% and 82.9% respectively versus 24.8%; p<0.001). Amongst the participants who had successful intubations, King Vision channeled and McGrath had shorter mean time-to-intubation compared to the King Vision non-channeled (41.3±20.3s and 38.5±18.7s respectively versus 53.8±23.8s, p<0.004;). There was no significant difference in the rate of successful intubation and mean time-to-intubation between King Vision channeled and McGrath. The King Vision channeled and McGrath video laryngoscopes demonstrated superior intubation success rates compared to King Vision non-channeled laryngoscope when used by junior doctors for intubating simulated difficult airway in an out-of-hospital setting. We postulated that the presence of a guidance channel in the King Vision channeled laryngoscope and the familiarity of the blade curvature and handling of the McGrath could have accounted for their improved intubation success rates.

## Introduction

Failure to secure the airway is an important cause of morbidity and mortality during resuscitations [1,2]. Video laryngoscopes have been consistently demonstrated to be critical in difficult

**Funding:** The affliated commercial company 'RS Anaesthesia & Intensive Care' was set up by last author R. C. H. Siew RS after completion of study and played no role in the design or conduct of the study. The rest of the authors do not receive salary nor funding from R. C. H. Siew's private anaesthesia practice. The company did not have any role in the study design, data collection and analysis, decision to publish, or preparation of the manuscript. The authors have declared that they did not receive any funding nor payment from any of the manufacturers of the laryngoscopes used in this study.

**Competing interests:** The affliated commercial company 'RS Anaesthesia & Intensive Care' was set up by last author R. C. H. Siew RS after completion of study and played no role in the design or conduct of the study. The rest of the authors do not receive salary nor funding from R. C. H. Siew's private anaesthesia practice. This does not alter our adherence to PLOS ONE policies on sharing data and materials.

or failed intubations [3–5] and have been incorporated into many difficult airway algorithms [6–8].

The earlier generation of video laryngoscopes were expensive, bulky, and required familiarization training [9]. This precluded widespread use of video laryngoscopes apart from specialised areas such as operating rooms and intensive care units [10]. Newer video laryngoscopes are now more portable, with some models incorporating additional features such as the channeled blade. The presence of this channel can help to reduce the need for complex manipulation of the tracheal tube during intubation [11–12]. Although there are many studies comparing the intubation efficacy of different video laryngoscopes, few studies have compared the success rate and ease of use of the channeled versus non-channeled video laryngoscopes when used by junior doctors in an out-of-hospital or military setting [12–14].

Our primary aim was to compare the rate of successful intubation of the channeled and non-channeled blade video laryngoscopes when used by junior doctors to intubate a simulated difficult airway. Secondary outcomes assessed included time-to-visualization of glottis, time-to-intubation and ease of use of the video laryngoscopes. We also investigated the impact of blade curvature of the non-channeled video laryngoscope blades on their intubating efficacies.

## Materials and methods

A crossover study was conducted from January 2016 to March 2017. Study was approved by local institutional research committee SAF Joint Medical Conference (Research) BB52/12-4 on 2 Aug 2015. Written consent was obtained from participants.

All junior doctors from the Medical Officer Cadet Course of the Singapore Armed Forces were considered for this study. Junior doctors were defined as post graduate year two or three medical officers with 10 or less previous intubation attempts in real-life patients. Exclusion criteria were medical officers with more than 10 previous intubation attempts in real-life patients, refusal to give written informed consent or non-physician medical officers.

In this study, we used the acute angled channeled and non-channeled blades of King Vision™ aBlade™ (King Systems, Noblesville, IN, USA) video laryngoscope, and the Macintosh styled blade of the McGRATH™ MAC (Aircraft Medical, Edinburgh, UK) video laryngoscope. We chose the King Vision video laryngoscope because of its cost, portability and design for use in patients with limited mouth opening ($\geq$18mm and $\geq$13mm for the size 3 channeled and non-channeled blades respectively [15]) and restricted neck movement. These are important considerations in the management of difficult airway in an out-of-hospital or military setting. We also compared King Vision non-channeled with McGrath video laryngoscopes as McGrath uses a Macintosh blade design with a curvature that is less angled than that of King Vision.

All participants had to watch a standardised training video on the laryngoscopy technique using all three video laryngoscopes. We played the training video for KingVision (02m:05s) first, before the Mcgrath teaching video (00m:25s). The participants then undergo a familiarization session with the SimMan® 3G advanced patient simulator manikin prior to the start of the simulation session. To minimize the effects of time and fatigue, all the sessions took place during office hours, and none of the participants were post-call/ sleep-fatigued when participating in the study.

The participants intubated with each of the three devices following a sequence based on a 3-period, 3-treatment crossover design (Table 1) to minimise the learning effect on subsequent intubation attempts [11,14,16]. All participants were allowed up to 90 seconds to achieve tracheal intubation in a standardised simulated difficult airway scenario for each device. In

**Table 1. Intubation sequence based on a 3-period, 3-treatment crossover design.**

| Sequence | Period 1 | Period 2 | Period 3 |
|---|---|---|---|
| A | McGrath | King Vision channeled | King Vision non-channeled |
| B | McGrath | King Vision non-channeled | King Vision channeled |
| C | King Vision channeled | King Vision non-channeled | McGrath |
| D | King Vision channeled | McGrath | King Vision non-channeled |
| E | King Vision non-channeled | King Vision channeled | McGrath |
| F | King Vision non-channeled | McGrath | King Vision channeled |

the event of an oesophageal intubation, the timer continued and participants were allowed repeated attempts at intubation, up to a maximum of 90 seconds.

Successful intubation was defined as the ability to achieve tracheal intubation within 90 seconds and was demonstrated with the inflation of the manikin's lungs with a manual resuscitator. Time-to-visualization was defined as the time from taking hold of the laryngoscope handle to the visualization of the glottis by the participant. Time-to-intubation was defined as the total time taken from holding the laryngoscope handle to the confirmation of tracheal tube position by inflation of the manikin's lungs. Upon completion of the simulation session, participants were asked to indicate their preferred device in an outfield setting and to rate the ease of use of the video laryngoscopes using a Likert 5-point scale (1 = very easy, 2 = easy, 3 = moderate, 4 = difficult, 5 = very difficult) in an anonymous survey.

A consistent difficult airway scenario was simulated by activating the tongue-oedema feature in the SimMan® 3G manikin and by placing a standard cervical collar for cervical immobilisation (Fig 1). The participants also had to intubate the manikin on the ground, simulating intubation in an out-of-hospital setting. All tracheal intubations were performed using the size 3 blade for each of the three video laryngoscopes and a standard size 7 Portex® tracheal tube in the SimMan® manikin. A pre-formed stylet was inserted into the Portex® tracheal tube for intubation with the King Vision non-channeled and McGrath laryngoscopes, while no stylet was used with King Vision channeled laryngoscopes.

A pilot study conducted earlier with 21 junior doctors showed a rate of successful intubation of 85% when using the McGrath in the SimMan® simulated difficult airway scenario. A power analysis was done to determine sample size based on this pilot study. We hypothesised that the King Vision channeled video laryngoscope will improve rate of successful intubation

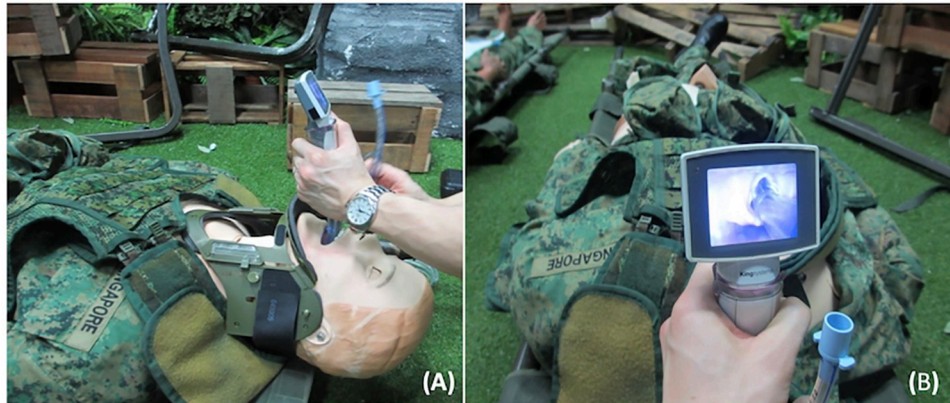

**Fig 1. (A) Demonstration on the use of the King Vision™ laryngoscope and (B) visualization of vocal cords in the SimMan® 3G manikin during the difficult airway scenario.**

**Table 2. Comparison of successful intubation and time-to-visualization between the King Vision channeled and non-channeled groups, and McGrath.** Values are number (proportion) or mean (standard deviation).

| | King Vision channeled (n = 105) | King Vision non-channeled (n = 105) | McGrath (n = 105) | p-value |
|---|---|---|---|---|
| Successful intubation | 90 (85.7%) | 26 (24.8%) | 87 (82.9%) | <0.001[i] |
| Time to visualization (seconds) | 12.1 (±7.3) | 10.3 (±12.9) | 13.9 (±11.7) | 0.054[ii] |

[i] p <0.001 for King Vision channeled versus non-channeled, and McGrath versus King Vision non-channeled; p = 0.569 for King Vision channeled versus McGrath

[ii] p = 0.651 for King Vision channeled versus non-channeled, p = 0.047 McGrath versus King Vision non-channeled; p = 0.704 for King Vision channeled versus McGrath

for junior doctors by 10% in the SimMan® simulated difficult airway scenario. Using a two-sided alpha level of 5% and statistical power of 80%, the required sample size to detect a 10% difference in the rate of successful intubation between the groups was calculated to be 77.

Statistical analysis was performed using SPSS Version 24 (SPSS Inc, Chicago IL, USA). The normal distribution of data was tested using the Shapiro-Wilk test. Comparison of success rates was analysed using Chi-squared tests. Analyses of continuous data were performed using one-way ANOVA (for parametric data) and independent-samples Kruskal-Wallis test (for non-parametric data) with Bonferroni correction. The ease of use of video laryngoscopes was analysed using Kruskal-Wallis test. A p-value of < 0.05 was considered significant.

## Results

A total of 105 junior doctors were recruited in this study. The median number of successful real-life intubations performed based on recall amongst participants was three. None of the participants had used the King Vision and McGrath laryngoscopes for real-life intubations prior to this study.

Rates of successful intubations were higher for the King Vision channeled and McGrath compared to the King Vision non-channeled video laryngoscopes (85.7% and 82.9% respectively versus 24.8%; p<0.001, Table 2). However, there was no significant difference in the rates of successful intubations between the King Vision channeled and McGrath. All participants managed to achieve visualization of the glottis, with a mean time-to-visualization of less than 15 seconds for all three devices.

Amongst participants who had successful intubations, King Vision channeled and McGrath had shorter mean time-to-intubation compared to the King Vision non-channeled (41.3±20.3s and 38.5±18.7s respectively versus 53.8±23.8s, p<0.004; Table 3). There was no significant difference in mean time-to-intubation between King Vision channeled and McGrath.

On a Likert 5-point scale for ease of use of video laryngoscopes, the median score for King Vision channeled and McGrath groups were 2 [interquartile range of 1–3], while the median score for King Vision non-channeled group was 4 [interquartile range of 3–5]. When asked to recommend a video laryngoscope amongst the three, 45.2% of participants recommended

**Table 3. Comparison of time-to-intubation for King Vision channeled and non-channeled groups, and McGrath.** Results are expressed as mean (standard deviation).

| | King Vision channeled (n = 90) | King Vision non-channeled (n = 26) | McGrath (n = 87) | p-value |
|---|---|---|---|---|
| Time to intubation (seconds) | 41.3 (±20.2) | 53.8 (±23.8) | 38.5 (±18.7) | 0.004[i] |

[i] p = 0.017 for King Vision channeled versus King Vision non-channeled; p = 0.002 for McGrath versus King Vision non-channeled; p = 1.000 for King Vision channeled versus McGrath

King Vision channeled, 1.5% recommended King Vision non-channeled, and 53.3% recommended McGrath.

## Discussion

Video laryngoscopes are designed to provide indirect visualization of the glottis. Acute angled blades, such as the ones in King Vision video laryngoscopes, enable the user to achieve better laryngeal view with less neck and laryngeal manipulation [8,17]. However, without a direct line of vision to the glottis, the passage of tracheal tube into the glottis require deft hand-eye coordination [9,17], which may pose a problem for junior doctors with limited experience in intubation [17–19]. The presence of a tracheal tube guidance channel can help to overcome the need for complex manipulation during intubation [11–12]. Our study showed that the King Vision channeled laryngoscopes had better success rates compared to the King Vision non-channeled laryngoscopes, despite having the same blade angulation and curvature. This is consistent with the study by Akihisa et al that having a guided channel for tracheal tube improved intubation performance compared to a non-channeled blade for novice operators [14].

In our study, we made the intubation process more realistic by simulating a difficult airway with tongue oedema, cervical immobilisation with standard cervical collar, and requiring participants to intubate the manikin on the ground, similar to an out-of-hospital or military setting. The importance of a guided channel for an acute angled blade is accentuated when intubation becomes more challenging. This was demonstrated in our study where the success rate for King Vision non-channeled was 24.8%, compared to 47.3% for King Vision non-channeled in the study by Akihisa et al, where a standard manikin airway was used. This observation was also echoed in a study by Okada et al, which yielded no significant difference in intubation success rates between the King Vision channeled and non-channeled groups when used by novice operators in standard manikin airways, but demonstrated higher intubation success rates with the King Vision channeled laryngoscopes when compared to the non-channeled ones (25/25 versus 18/25, p = 0.004) when intubation was performed on a manikin with ongoing chest-compressions [12].

McGrath also performed better than the King Vision non-channeled laryngoscopes in terms of intubation success rates. While the McGrath did not have a channeled blade, it had the advantage of a blade curvature similar to the standard Macintosh blade, which the junior doctors were more familiar with. Other studies had also observed that junior doctors were able to translate their intubating skills if the curvature of the video laryngoscope blade was similar to the standard Macintosh blade [20–21]. We postulated that it was possible that our participants found it easier to acquaint themselves with the McGrath and achieved better success rates than with the King Vision non-channeled laryngoscope because of their familiarity with the McGrath blade curvature and handling.

When designing our study, we believe that it was important to break down the intubation process into two components: (1) achieving glottis visualization followed by (2) manipulation of tracheal tube past the glottis. Many studies agreed that the improved laryngeal view associated with video laryngoscope use did not necessarily lead to intubation success [3,13,21–22]. In our study, all 3 video laryngoscopes achieved a mean time-to-visualization of less than 15s. However, the King Vision non-channeled laryngoscope had the longest time-to-intubation. To improve intubation success rates with the King Vision non-channeled laryngoscope, we postulated that operators should be well-trained in the use of stylets, which some considered to be mandatory for intubation with angulated blades without a guiding channel [17,23]. They should be adept at shaping the stylets optimally for intubation, and possess good hand-eye

coordination for manipulation of the tracheal tube when the oro-pharyngeal axis and the pharyngo-glotto-tracheal axis are not aligned to provide direct glottic view [9,13,17]. It will be interesting to determine whether the time to intubation using video laryngoscopes with acute angulated blades without a guided channel, such as the King Vision non-channeled video laryngoscope, differs between expert video laryngoscopists and less experienced medical personnel when intubating difficult airways.

There was no significant difference in terms of intubation success rates between the King Vision channeled and McGrath laryngoscopes. There were also no significant differences between King Vision channeled and McGrath in terms of time-to-visualization and time-to-intubation. Participants also rated both King Vision channeled and McGrath to be "easy to use".

Other studies comparing various video laryngoscopes have also found that a variety of video laryngoscope are potentially appropriate for novice operators [4,11,13,24]. McGrath, with its standard Macintosh blade, offered the advantage of improved laryngeal view for operators who are familiar with its use. In a randomised controlled trial done by B. Alvis et al, experienced laryngoscopists with limited experience in video laryngoscopes were more successful and intubated faster with McGrath compared to King Vision channeled laryngoscopes [20]. On the other hand, novice operators, who might not be as adept with the technique of direct laryngoscopy, might prefer the King Vision channeled laryngoscopes because of its easy manipulation past the tongue for glottis visualization and a channel for tracheal guidance into the glottis [25–28]. King Vision channeled and non-channeled laryngoscopes have also been described for use as an alternative to awake fibreoptic intubation in patients with limited mouth opening and/or neck movement [15,29].

There were several limitations to our study. Firstly, while the SimMan® manikins used in this study are high fidelity advanced patient simulators, intubating a manikin is different from intubating a real patient. Hence the results might not be directly applicable to real-life patients. Secondly, while the SimMan® allowed for standardisation of the difficult airway scenario for our participants, this was limited to cervical immobilisation and tongue oedema. We recognised that difficult airways scenarios in actual trauma or collapsed patients may include other variables such as blood and secretions in the oral cavity, anatomic variations and/or facial injuries, which are beyond the scope of this study. Demographic differences, which were not measured, could have made a difference to the ease of intubation. Taller, heavier participants might find it difficult to kneel down on the ground to intubate compared to smaller participants.

## Supporting information

**S1 Fig.**
(PNG)

**S1 File. Data Set.**
(SAV)

**S1 Table.**
(DOCX)

**S2 Table.**
(DOCX)

**S3 Table.**
(DOCX)

## Acknowledgments

We are grateful to Professor Lee Tat Leang from the National University Hospital Department of Anaesthesia for his review of this manuscript and Ms Liu Wei Ling for her advice on the statistical analysis.

## Author Contributions

**Conceptualization:** Shi Hao Chew, Jonathan Zhao Min Lim, Benjamin Zhao Bin Chin, Raymond Chern Hwee Siew.

**Data curation:** Shi Hao Chew, Jonathan Zhao Min Lim, Benjamin Zhao Bin Chin, Raymond Chern Hwee Siew.

**Formal analysis:** Shi Hao Chew, Jia Xin Chan.

**Investigation:** Shi Hao Chew, Raymond Chern Hwee Siew.

**Methodology:** Shi Hao Chew, Jonathan Zhao Min Lim, Benjamin Zhao Bin Chin, Raymond Chern Hwee Siew.

**Project administration:** Shi Hao Chew, Jonathan Zhao Min Lim, Benjamin Zhao Bin Chin, Raymond Chern Hwee Siew.

**Resources:** Jia Xin Chan.

**Software:** Jia Xin Chan.

**Supervision:** Raymond Chern Hwee Siew.

**Validation:** Jonathan Zhao Min Lim, Jia Xin Chan, Raymond Chern Hwee Siew.

**Writing – original draft:** Shi Hao Chew.

**Writing – review & editing:** Shi Hao Chew, Jonathan Zhao Min Lim, Benjamin Zhao Bin Chin, Jia Xin Chan, Raymond Chern Hwee Siew.

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
