## [Decision Letter · Decision Letter 0]

13 Aug 2019

PONE-D-19-18575

Intubation with channelled versus non-channelled video laryngoscopes in simulated difficult airway by junior doctors in an out-of-hospital setting: A crossover manikin study.

PLOS ONE

Dear Dr. Chew 

Thank you for submitting your manuscript to PLOS ONE. After careful consideration, we feel that it has merit but does not fully meet PLOS ONE’s publication criteria as it currently stands. Therefore, we invite you to submit a revised version of the manuscript that addresses the points raised during the review process.

I would appreciate to clarify the clinical importance of your study and in to response to the reviewers queries specially those of the second reviewer.  

We would appreciate receiving your revised manuscript by Sep 20 2019 11:59PM. To enhance the reproducibility of your results, we recommend that if applicable you deposit your laboratory protocols in protocols.io, where a protocol can be assigned its own identifier (DOI) such that it can be cited independently in the future. For instructions see: http://journals.plos.org/plosone/s/submission-guidelines#loc-laboratory-protocols

We look forward to receiving your revised manuscript.

Kind regards,

Ehab Farag, MD FRCA FASA

Academic Editor

PLOS ONE

Journal Requirements:

[The author(s) received no specific funding for this work.].   

We note that one or more of the authors are employed by a commercial company: 'RS Anaesthesia & Intensive Care,'.

Reviewers' comments:

Reviewer's Responses to Questions

**Comments to the Author**

1. Is the manuscript technically sound, and do the data support the conclusions?

Reviewer #1: Yes

Reviewer #2: Partly

2. Has the statistical analysis been performed appropriately and rigorously? 

Reviewer #1: I Don't Know

Reviewer #2: Yes

3. Have the authors made all data underlying the findings in their manuscript fully available?

Reviewer #1: Yes

Reviewer #2: Yes

4. Is the manuscript presented in an intelligible fashion and written in standard English?

Reviewer #1: Yes

Reviewer #2: Yes

5. Review Comments to the Author

Reviewer #1: This is a useful contribution to the airway literature, even though it is a mannequin study with the usual caveats that apply to mannequin airway studies. I could identify no methodological flaws in my reading of the manuscript. While I am not qualified to assess the statistical methodology used in the study, it appears to be sound. My only real concern is that the language used is in some need of polishing. In addition, the manuscript used British rather than American medical spelling.

Reviewer #2: The authors have studied % success rate of intubation of junior doctors with 3 videolaryngoscopic devices (King Vision channeled, King Vision non channeled, McGrath) in a standardized manikin under simulated difficult airway conditions. It is a well conducted study. Certain concerns need to be addressed:

1. Their conclusion is not in line with their results. While the participants recommended KV channeled laryngoscope 45.2% of the times and McGrath 53.3% of the times, the reasons for this were never explored. Instead, the authors assume or hypothesize that presence of the channel for the KV scope and easy handling of McGrath were responsible for these choices. The study does not explore reasons of why did the participants felt that KV channeled and McGrath scopes were preferable. Thus the statement, on page 2, lines 34 -36: "The presence of a guidance

channel in the King Vision channeled laryngoscope and the familiarity of the blade

curvature and handling of the McGrath are important factors, which contributed to the

ease of use for junior doctors." is invalid. This needs to be recognized and changed as this is the authors interpretation and not evidence based.

2. On page 4 lines 89 to 93: "All participants were allowed up to 90 seconds to achieve tracheal intubation in a standardized simulated difficult airway scenario for each device. In the event of an esophageal intubation, the timer will continue and participants were allowed repeated attempts at intubation, up to a maximum of 90seconds." Instead of the timer will continue, it is best reworded as the timer continued. Additionally, I am curious to why the authors did not record intubation attempts over the allowed 90 second period. Intubation attempts often worsen edema in real patients, and this is a meaningful outcome. In addition to success of intubation as defined by the authors, it would be meaningful to know which device was associated with least number of intubation attempts.

3. The Methodology needs more clarification. How many junior doctors were assigned to each sequence? Were effects of time and fatigue taken into account? Did all the doctors participate simultaneously at the same time (For example: 105 doctors intubated between 9 and 11am on 6 standardized manikins) or did they participate in discrete groups at different times (20 doctors at a time on a single manikin at different time periods such as 9 to 11 am , then noon to 2 pm etc)? Were all participants well rested or were some participating after being on call or coming off their regular work shifts? This may affect retention of video content that was played prior to having the participants intubate. Additionally, length of video is important. Retention is best for videos that are under 6 minutes. Reference: Philip J. Guo, Juho Kim, and Rob Rubin. 2014. How video production affects student engagement: an empirical study of MOOC videos. In Proceedings of the first ACM conference on Learning @ scale conference (L@S '14). ACM, New York, NY, USA, 41-50. DOI: https://doi.org/10.1145/2556325.2566239

Thus, if McGrath gets talked about in the early part of the video, participants may remember more about using McGrath than King Vision.

3. Demographic differences are important. Taller participants and heavier participants may find it difficult to kneel on the ground and intubate. If the authors have this information, it may be valuable to see if this affected success of intubation.

4. Time to visualization and Time to intubation make for important statistical outcomes but clinically the difference may not be relevant. For example, time to visualization for KV non channeled scope was 10 seconds vs 12 and 14 s for KV channeled and McGrath scopes. Would a 2 or 4 second difference be important clinically?

5. In Table 2, authors list the number of successful intubations for KV channeled, non channeled and McGrath devices as 90, 26 and 87. This changes to 91, 28 and 88 in Table 3. Could the authors shed more light on this?

6. PLOS authors have the option to publish the peer review history of their article (what does this mean?). If published, this will include your full peer review and any attached files.

Reviewer #1: No

Reviewer #2: Yes: Sandeep Khanna

---

## [Author Response · Author response to Decision Letter 0]

26 Sep 2019

Dear Ehab Farag (Academic Editor PLOS ONE),

Thank you for the reply. Our responses are as follows:

1. We will adhere to the formatting guidelines, and update the manuscript accordingly

2. Financial Disclosure Statement. The author(s) received no specific funding for this work. The affliated commercial company 'RS Anaesthesia & Intensive Care' was set up by last author R. C. H. Siew RS after completion of study and played no role in the design or conduct of the study. The rest of the authors do not receive salary nor funding from R. C. H. Siew’s private anaesthesia practice. The company did not have any role in the study design, data collection and analysis, decision to publish, or preparation of the manuscript.

3. Competing Interests. There are no competing interest. The affliated commercial company 'RS Anaesthesia & Intensive Care' was set up by last author R. C. H. Siew RS after completion of study and played no role in the design or conduct of the study. The rest of the authors do not receive salary nor funding from R. C. H. Siew’s private anaesthesia practice. This does not alter our adherence to PLOS ONE policies on sharing data and materials.

4. Review Comments to the Author

Reviewer #1: This is a useful contribution to the airway literature, even though it is a mannequin study with the usual caveats that apply to mannequin airway studies. I could identify no methodological flaws in my reading of the manuscript. While I am not qualified to assess the statistical methodology used in the study, it appears to be sound. My only real concern is that the language used is in some need of polishing. In addition, the manuscript used British rather than American medical spelling.

We will re-work the language, and align the manuscript to use American medical spelling.

Reviewer #2: The authors have studied % success rate of intubation of junior doctors with 3 videolaryngoscopic devices (King Vision channeled, King Vision non channeled, McGrath) in a standardized manikin under simulated difficult airway conditions. It is a well conducted study. Certain concerns need to be addressed:

1. Their conclusion is not in line with their results. While the participants recommended KV channeled laryngoscope 45.2% of the times and McGrath 53.3% of the times, the reasons for this were never explored. Instead, the authors assume or hypothesize that presence of the channel for the KV scope and easy handling of McGrath were responsible for these choices. The study does not explore reasons of why did the participants felt that KV channeled and McGrath scopes were preferable. Thus the statement, on page 2, lines 34 -36: "The presence of a guidance

channel in the King Vision channeled laryngoscope and the familiarity of the blade

curvature and handling of the McGrath are important factors, which contributed to the

ease of use for junior doctors." is invalid. This needs to be recognized and changed as this is the authors interpretation and not evidence based.

We will adjust our statement to reflect that this is the authors’ interpretation of the results.

2. On page 4 lines 89 to 93: "All participants were allowed up to 90 seconds to achieve tracheal intubation in a standardized simulated difficult airway scenario for each device. In the event of an esophageal intubation, the timer will continue and participants were allowed repeated attempts at intubation, up to a maximum of 90seconds." Instead of the timer will continue, it is best reworded as the timer continued. Additionally, I am curious to why the authors did not record intubation attempts over the allowed 90 second period. Intubation attempts often worsen edema in real patients, and this is a meaningful outcome. In addition to success of intubation as defined by the authors, it would be meaningful to know which device was associated with least number of intubation attempts.

We did not record intubation attempts over the allowed 90 second period, as timing is of essence during emergent airway intubation. This is in line with the ASA Difficult Airway Guidelines, which discourage repeated attempts at intubation for the same reason cited by the reviewer – because repeated intubation attempts often worsen oedema in real patients.

3. The Methodology needs more clarification. How many junior doctors were assigned to each sequence? Were effects of time and fatigue taken into account? Did all the doctors participate simultaneously at the same time (For example: 105 doctors intubated between 9 and 11am on 6 standardized manikins) or did they participate in discrete groups at different times (20 doctors at a time on a single manikin at different time periods such as 9 to 11 am , then noon to 2 pm etc)? Were all participants well rested or were some participating after being on call or coming off their regular work shifts? This may affect retention of video content that was played prior to having the participants intubate. Additionally, length of video is important. Retention is best for videos that are under 6 minutes. Reference: Philip J. Guo, Juho Kim, and Rob Rubin. 2014. How video production affects student engagement: an empirical study of MOOC videos. In Proceedings of the first ACM conference on Learning @ scale conference (L@S '14). ACM, New York, NY, USA, 41-50. DOI: https://doi.org/10.1145/2556325.2566239

Thus, if McGrath gets talked about in the early part of the video, participants may remember more about using McGrath than King Vision.

Thank you for the input. We will add more clarity in our methodology. About 10-25 subjects were tested in discrete groups at different times over multiple sessions. All of these sessions took place during office hours, and none of the participants were post-call/ or under the effects of sleep fatigue when participating in the study. 

3. Demographic differences are important. Taller participants and heavier participants may find it difficult to kneel on the ground and intubate. If the authors have this information, it may be valuable to see if this affected success of intubation.

The reviewer raised a valid point that demographic variations may affect ease of intubation. Our study group did not record the height and weight of participants. While it may be true that taller and heavier participants may find it more difficult to kneel on the ground and intubate, the research was scoped to study how curvature and presence of channelled video laryngoscopy blade. In real life, we do not discriminate doctors based on their height and weight during deployment. 

4. Time to visualization and Time to intubation make for important statistical outcomes but clinically the difference may not be relevant. For example, time to visualization for KV non channeled scope was 10 seconds vs 12 and 14 s for KV channeled and McGrath scopes. Would a 2 or 4 second difference be important clinically?

The differences in Time to Visualisation (TTV) are not as clinically important as differences in Time to Intubation, which may have implications such as hypoxic brain injury. In reality, a delay of 2-4s is unlikely to have significant clinical impact. 

5. In Table 2, authors list the number of successful intubations for KV channeled, non channeled and McGrath devices as 90, 26 and 87. This changes to 91, 28 and 88 in Table 3. Could the authors shed more light on this?

The correct values should be n= 90, 26, and 87. I ran the statistics again, and realised that this error occurred when we input missing values (failed intubations) as 99.0s. This led to additional n values during analysis of TTI. We have corrected this, and have included the raw analysis from SPSS below.

---

## [Editor Report · Decision Letter 1]

4 Oct 2019

Intubation with channelled versus non-channelled video laryngoscopes in simulated difficult airway by junior doctors in an out-of-hospital setting: A crossover manikin study.

PONE-D-19-18575R1

Dear Dr. Shi Hao Chew

We are pleased to inform you that your manuscript has been judged scientifically suitable for publication and will be formally accepted for publication once it complies with all outstanding technical requirements.

With kind regards,

Ehab Farag, MD FRCA FASA

Academic Editor

PLOS ONE
---

## [Editor Report · Acceptance letter]

15 Oct 2019

PONE-D-19-18575R1 

Intubation with channeled versus non-channeled video laryngoscopes in simulated difficult airway by junior doctors in an out-of-hospital setting: A crossover manikin study. 

Dear Dr. Chew:

I am pleased to inform you that your manuscript has been deemed suitable for publication in PLOS ONE. Congratulations! Your manuscript is now with our production department. 

With kind regards,

on behalf of

Dr. Ehab Farag 

Academic Editor

PLOS ONE